# An Exploratory Study of Refining TNM-8 M1 Categories and Prognostic Subgroups Using Plasma EBV DNA for Previously Untreated De Novo Metastatic Nasopharyngeal Carcinoma

**DOI:** 10.3390/cancers14081923

**Published:** 2022-04-11

**Authors:** Sik-Kwan Chan, Brian O’Sullivan, Shao Hui Huang, Tin-Ching Chau, Ka-On Lam, Sum-Yin Chan, Chi-Chung Tong, Varut Vardhanabhuti, Dora Lai-Wan Kwong, Chor-Yi Ng, To-Wai Leung, Mai-Yee Luk, Anne Wing-Mui Lee, Horace Cheuk-Wai Choi, Victor Ho-Fun Lee

**Affiliations:** 1Department of Clinical Oncology, LKS Faculty of Medicine, The University of Hong Kong, Hong Kong, China; csk01@connect.hku.hk (S.-K.C.); chautc@hku.hk (T.-C.C.); lamkaon@hku.hk (K.-O.L.); csy023@ha.org.hk (S.-Y.C.); tccz01@ha.org.hk (C.-C.T.); dlwkwong@hku.hk (D.L.-W.K.); ncy515@ha.org.hk (C.-Y.N.); leungtw@ha.org.hk (T.-W.L.); lmyz01@ha.org.hk (M.-Y.L.); awmlee@hku.hk (A.W.-M.L.); 2Department of Radiation Oncology, Princess Margaret Cancer Centre, University of Toronto, Toronto, ON M5G 2C1, Canada; brian.osullivan@rmp.uhn.ca (B.O.); shaohui.huang@rmp.uhn.ca (S.H.H.); 3Clinical Oncology Center, The University of Hong Kong-Shenzhen Hospital, Shenzhen 518053, China; 4Department of Diagnostic Radiology, LKS Faculty of Medicine, The University of Hong Kong, Hong Kong, China; varv@hku.hk; 5School of Public Health, LKS Faculty of Medicine, The University of Hong Kong, Hong Kong, China

**Keywords:** nasopharyngeal carcinoma, metastatic, M1 categories, plasma EBV DNA, recursive partitioning analysis, TNM

## Abstract

**Simple Summary:**

Patients with de novo metastatic (M1) nasopharyngeal carcinoma (NPC) at presentation is a heterogeneous group of the population who have a diverse range of survival. However, the current TNM-8 grouping of all these patients into the M1 category is not able to identify the survival differences among them. We aimed to segregate survival for de novo M1 NPC by anatomic characteristics and pre-treatment plasma Epstein–Barr virus (EBV) DNA, respectively. We first proposed a potential M1 subdivision with anatomic factors for de novo M1 NPC, which can be in general applied in different geographical regions. Further recursive-partitioning analysis (RPA)-derived prognostic groupings with plasma EBV DNA at 2500 copies/mL performed better in survival prediction and risk stratification, resulting in a potentially more precise and personalized treatment. Further external validation of our proposed M1 stage subdivisions in other institutions is highly awaited.

**Abstract:**

(1) Background: NPC patients with de novo distant metastasis appears to be a heterogeneous group who demonstrate a wide range of survival, as suggested by growing evidence. Nevertheless, the current 8th edition of TNM staging (TNM-8) grouping all these patients into the M1 category is not able to identify their survival differences. We sought to identify any anatomic and non-anatomic subgroups in this study. (2) Methods: Sixty-nine patients with treatment-naive de novo M1 NPC (training cohort) were prospectively recruited from 2007 to 2018. We performed univariable and multivariable analyses (UVA and MVA) to explore anatomic distant metastasis factors, which were significantly prognostic of overall survival (OS). Recursive partitioning analysis (RPA) with the incorporation of significant factors from MVA was then performed to derive a new set of RPA stage groups with OS segregation (Set 1 Anatomic-RPA stage groups); another run of MVA was performed with the addition of pre-treatment plasma EBV DNA. A second-round RPA with significant prognostic factors of OS identified in this round of MVA was performed again to derive another set of stage groups (Set 2 Prognostic-RPA stage groups). Both sets were then validated externally with an independent validation cohort of 67 patients with distant relapses of their initially non-metastatic NPC (rM1) after radical treatment. The performance of models in survival segregation was evaluated by the Akaike information criterion (AIC) and concordance index (C-index) under 1000 bootstrapping samples for the validation cohort; (3) Results: The 3-year OS and median follow-up in the training cohort were 36.0% and 17.8 months, respectively. Co-existence of liver-bone metastases was the only significant prognostic factor of OS in the first round UVA and MVA. Set 1 RPA based on anatomic factors that subdivide the M1 category into two groups: M1a (absence of co-existing liver-bone metastases; median OS 28.1 months) and M1b (co-existing liver-bone metastases; median OS 19.2 months, *p* = 0.023). When pre-treatment plasma EBV DNA was also added, it became the only significant prognostic factor in UVA (*p* = 0.001) and MVA (*p* = 0.015), while co-existing liver-bone metastases was only significant in UVA. Set 2 RPA with the incorporation of pre-treatment plasma EBV DNA yielded good segregation (M1a: EBV DNA ≤ 2500 copies/mL and M1b: EBV DNA > 2500 copies/mL; median OS 44.2 and 19.7 months, respectively, *p* < 0.001). Set 2 Prognostic-RPA groups (AIC: 228.1 [95% CI: 194.8–251.8] is superior to Set 1 Anatomic-RPA groups (AIC: 278.5 [254.6–301.2]) in the OS prediction (*p* < 0.001). Set 2 RPA groups (C-index 0.59 [95% CI: 0.54–0.67]) also performed better prediction agreement in the validation cohort (vs. Set 1: C-index 0.47 [95% CI: 0.41–0.53]) (*p* < 0.001); (4) Conclusions: Our Anatomic-RPA stage groups yielded good segregation for de novo M1 NPC, and prognostication was further improved by incorporating plasma EBV DNA. These new RPA stage groups for M1 NPC can be applied to countries/regions regardless of whether reliable and sensitive plasma EBV DNA assays are available or not.

## 1. Introduction

The American Joint Committee on Cancer/Union for International Cancer Control (AJCC/UICC) TNM staging classification is the internationally recognized lingua franca to describe tumor extent [1,2,3,4]. In view of the current diagnostic and therapeutic advances, the T- and N-categories of non-metastatic nasopharyngeal carcinoma (NPC) have been revised in its 8th edition (TNM-8), while no change has been made in de novo metastatic NPC, which are all denoted by an inclusive “M1” category [1,2,3,4]. Nevertheless, recent evidence suggested that NPC patients with distant metastasis (DM) at presentation appear to be a heterogeneous group with a wide range of survival. Previous retrospective studies identified some prognostic factors of overall survival (OS) for metastatic NPC, including age, location, and the number of metastatic lesions and treatment strategies [5,6,7,8,9,10,11,12,13,14,15,16,17,18,19,20]. Persistent disease control was observed in bone-only oligometastatic NPC [21]. Our recently published pooled analysis demonstrated that the co-existence of liver and bone metastases is a significantly worse prognostic factor of OS for de novo M1 NPC [22]. These highlighted the importance of the M1 category subdivision to improve survival prediction and patient stratification for subsequent personalized treatment.

Besides anatomic DM factors, certain biomarkers, especially plasma Epstein–Barr virus (EBV) DNA might also facilitate risk stratification. Plasma EBV DNA has played an important role in NPC screening, prognostication, and surveillance [23,24,25,26,27,28,29,30,31,32]. While international multicenter harmonization of EBV DNA assays is still underway [33], the EBV DNA assay first established by Lo et al. in Hong Kong and used by us has been so far the most reliable and sensitive, which is considered the gold standard in endemic regions [23,30,33]. Though the prognostic role of pre-treatment plasma EBV DNA has been widely validated and accepted in non-metastatic NPC, its role in de novo M1 NPC, in particular in staging, remains unclear.

Therefore, further to our recent pooled analysis [22], we conducted a prospective study to evaluate whether certain anatomic DM characters can stratify de novo M1 NPC by multivariable analysis (MVA) and recursive partitioning analysis (RPA) and the inclusion of pre-treatment plasma EBV DNA to derive prognostic risk groups would further improve prognostic stratification.

## 2. Materials and Methods

### 2.1. Patient Recruitment

Patients with newly diagnosed and histologically proven NPC in our single institution were recruited from 2007 onwards. Sixty-nine patients with treatment-naive and histologically or radiologically confirmed distant metastasis at the time of diagnosis (de novo M1 NPC) and without other malignancies formed the training cohort. A validation cohort was formed from another 67 out of 518 prospectively recruited patients who developed distant metastases (rM1) following radical intensity-modulated radiation therapy (IMRT) +/– concurrent with or without additional adjuvant/induction chemotherapy for their originally non-metastatic NPC, as previously reported by us [33]. The local institutional review board (Institutional Review Board of The University of Hong Kong/Hospital Authority Hong Kong West Cluster, Hong Kong, China) approved (25 June 2008) the study (IRB reference number UW 08-231, UW 10-018 and UW 12-153), which was compiled with the guidelines of Declaration of Helsinki and Good Clinical Practice and REMARK recommendations. The study was registered with Clinicaltrials.gov, accessed on 12 March 2022 (NCT02476669). The study protocol and treatment details have been described in our previous publications [30,33].

### 2.2. Diagnosis and Treatment

After informed consent, complete pre-treatment investigations were undertaken, as detailed in Appendix A. Independent review of all images of positron emission tomography (PET-CT) and magnetic resonance imaging (MRI) and staging were conducted by two radiologists who specialize in head and neck radiology separately and blindly. Re-staging according to TNM-8 was conducted for this study. Any discrepancies that arose were settled by consensus.

Sixty-nine patients with de novo M1 NPC at diagnosis were prospectively recruited as the training cohort between 2007 and 2016. Of them, 64 (92.8%) received initial systemic chemotherapy with gemcitabine (G) 1000 mg/m^2^ on days 1 and 8 and cisplatin (P) 100 mg/m^2^ on day 1, every 3 weeks. A PET-CT scan was performed after 3 cycles of GP for response assessment based on RECIST 1.1. Plasma EBV DNA was also measured after 3 and 6 cycles of GP as part of the tumor response evaluation. Forty-eight (69.6%) patients without a progressive disease (as defined by RECIST 1.1) after the initial 3 cycles of chemotherapy continued the same regimen for up to 6 cycles, followed by consolidative IMRT (details in the Appendix A). Patients who had satisfactory performance status and physical condition also received concurrent P 100 mg/m^2^ or carboplatin (area under the curve [AUC] = 5) (if estimated creatinine clearance was <60 mL/minute) on day 1, 22, and 43 of chemoradiation, as the discretion of the treating oncologists. For patients with oligometastases at the time of diagnosis or disease recurrence following first-line systemic chemotherapy and consolidative IMRT to the nasopharynx and the neck, stereotactic body radiation therapy would be delivered to these oligometastases.

The treatment protocol of the patients in the validation cohort was similar to that for patients in the training cohort, except that consolidative IMRT was not given since they had had received radical IMRT +/– chemotherapy for their non-metastatic disease previously [30,33].

### 2.3. Plasma EBV DNA Measurement

Pre-treament plasma EBV DNA of the study patients in the training cohort and plasma EBV DNA at the time of distant relapse of the patients in the validation cohort were measured with the assay method described by Lo et al. and previously by us. Detailed determination of plasma EBV DNA is described in the Appendix A [23,30,33].

### 2.4. Treatment Monitoring

All patients were subject to fiberoptic nasoendoscopy 8 weeks after IMRT was completed and nasopharyngeal biopsies were performed if there were still suspicious residual tumors. As our routine practice, plasma EBV DNA titers were also measured again at 8 weeks after IMRT [30,33]. Patients with persistent tumor 3 months after the completion of IMRT would be given local salvage treatment, e.g., SBRT, surgery, RT. Surveillance clinical follow-up were performed every two to three months afterwards. Scans, including CT and/or MRI of the head and neck, thorax, and abdomen, were performed at a 3–4 months interval. PET-CT scan would be done if there was a clinical suspicion of relapse. Plasma EBV DNA of the patients was also evaluated every 3 months, at the time of further progressive disease and during further palliative systemic treatment until when all systemic treatment ended or when patients opted for palliative care only.

### 2.5. Statistical Analysis

We compared the demographic and clinical characteristics of the patients in the training and validation cohorts by Mann–Whitney U-tests for continuous variables and Chi-square tests for discrete categorical variables. The study endpoint was OS, the time that passed from the date of diagnosis of de novo M1 NPC for the training cohort (or the date of distant relapse for the rM1 validation cohort) to the date of death from any cause for both cohorts.

#### 2.5.1. Multivariable Analysis and Recursive Partitioning Analysis

We first identified the significant prognostic factors of OS in the training cohort, including anatomic DM factors as covariate by univariable analysis (UVA) and MVA, respectively. We then performed the first run of RPA incorporating the significant prognostic factors from MVA, so as to derive Set 1–Anatomic-RPA groups.

We further included all the above covariates and pre-treatment plasma EBV DNA (in log scale) into UVA and MVA again to investigate if plasma EBV DNA was prognostic of OS or not. Subsequently, based on the demographic and anatomic DM factors together with the addition of pre-treatment EBV DNA, we performed another RPA again to derive another set of M1 subgroups (Set 2–Prognostic-RPA groups). The optimal cutoff value of the plasma EBV DNA for the risk stratification was determined in the splitting process in the RPA of this round.

#### 2.5.2. Model Comparison and Validation

Both Set 1 and Set 2 were validated internally by 3-fold, 5-fold, and 10-fold, respectively with 1000 bootstraps. They were then compared with each other based on the Akaike information criterion (AIC) with OS as the survival endpoint. A lower AIC indicates a better set for survival prediction. The details and rationale of our RPA were described in Appendix A.

These two sets of RPA groups generated from the training cohort were then applied and validated in the validation cohort. To evaluate and compare the performance of Set 1 Anatomic-RPA groups and Set 2 Prognostic-RPA groups on OS prediction in the validation cohort, their concordance indices (C-index) were calculated with another 1000 bootstrap samples. The value of the C-index ranges between 0 and 1; a greater value of the C-index indicates higher discrimination between low and high-risk patient subgroups.

Finally, survival outcomes among subgroups in Set 1 and Set 2 were compared by Kaplan–Meier methods and log-rank tests.

All statistical analyses were performed using R version 3.5.3. *p* < 0.05 (two-sided) indicated statistical significance.

### 2.6. Literature Search

We also performed a comprehensive systematic review, which complied with the guidelines of the “Preferred Reporting Items for Systematic Review and Meta-analysis” (PRISMA) guidelines for publications, which included metastatic NPC treated with chemotherapy as initial systemic treatment. Significant databases were employed and searched from 1990 to the present day, with the most recent search carried out on 31 December 2021. The review was also registered on PROSPERO (registration number CRD42019147994). Keywords and medical sub-heading (MeSH) terms used in the search strategy covered the following concepts: nasopharyngeal, metastatic, EBV DNA, chemotherapy and prospective. The keywords and MeSH terms within each concept were then separated by the Boolean operator “AND”. The details and results of the literature search are provided in Appendix A. In summary, none of the publications conducted as a prospective study fulfilled our selection criteria in which plasma EBV DNA was proven as a prognostic biomarker in stage segregation of previously treatment-naïve de novo metastatic disease treated with initial chemotherapy followed by consolidative IMRT with or without additional local ablative treatment.

## 3. Results

### 3.1. Patient Characteristics

A flowchart of the study is presented in Figure 1. From January 2007 to September 2018, 69 consecutive patients (training cohort) with treatment-naive de novo M1 NPC were recruited prospectively and analyzed with their depositions shown (Table 1). Metastasis in bone, liver, lung as well as distant lymph nodes (LN) were found in 51 (73.9%), 23 (33.3%), 15 (21.7%) and 20 (29.0%) patients respectively. Thirty-three (44.9%) patients had multiple metastatic sites, with 16 (23.2%) of them suffering from concurrent liver and bone metastases (Figure 2). Fifty-six (81.2%) patients had their NPC controlled (i.e., without progressive disease) after the initial 6 cycles of GP chemotherapy. Consolidative IMRT was given to 48 (69.6%) of them. All patients with ≤3 oligometastases (4/69) (5.8%) at the time of diagnosis received SBRT after first-line chemotherapy and the subsequent consolidative IMRT. Eight (11.6%) patients, despite non-progression to initial chemotherapy, refused consolidative IMRT and SBRT due to personal decisions and had regular surveillance and follow-up until further progressive disease.

### 3.2. UVA and MVA for Prognostic Factors of OS

To identify prognostic factors, UVA and MVA were first performed for both the training and the validation cohorts. When only patient demographic and anatomic DM factors were included, co-existing liver-bone metastases was the only significant prognostic factor of OS in UVA (*p* = 0.023) and MVA (*p* = 0.023) (Table 2).

### 3.3. Establishment of M1 Subgroups by RPA

#### 3.3.1. Set 1 Anatomic-RPA Groups

Set 1 Anatomic RPA-groups by incorporating co-existence of liver-bone metastases as the only significant prognostic factor of OS from MVA into RPA could significantly segregate M1 into two categories: M1a–absence of co-existing liver-bone metastases, and M1b–co-existence of liver-bone metastases in the training cohort (Appendix A). Sixteen (23.2%) patients were presented with the involvement of both liver and bone metastases; their dispositions are listed in Table 3. Improved OS (median: 28.1 months, 3-year rate: 41.8%, 95% confidence interval [CI]: [25.8–57.8%]) was demonstrated in patients M1a when compared with their counterparts in M1b (p = 0.023) (Figure 3a, Table 3).

#### 3.3.2. Set 2 Prognostic-RPA Groups

We then added pre-treatment plasma EBV DNA, together with all the demographic and anatomic DM factors covariates and performed UVA and MVA again for the training cohort. When pre-treatment plasma EBV DNA was also added, it became the only significant prognostic factor in UVA (*p* = 0.001) and MVA (*p* = 0.015), while co-existing liver-bone metastases were only significant in UVA (Table 4).

Prognostic-RPA groups, after combining pre-treatment plasma EBV DNA as the only prognostic factor of OS in a separate RPA analysis for the training cohort (please refer to Appendix A on how the plasma EBV DNA cutoff was derived in this RPA), provided further prognostic groups (M1a (pre-treatment plasma EBV DNA ≤2500 copies/mL) and M1b (pre-treatment plasma EBV DNA >2500 copies/mL)) (Appendix A). Their corresponding 3-year OS were 74.4% (95% CI: 50.5–93.3%) and 17.1% (95% CI: 3.8–30.5%), respectively (*p* < 0.001) (Figure 4a and Table 5). The median OS was 44.2 months vs. 19.7 months.

A comparison of performance between Set 1 and Set 2 on predicting OS was also performed. After 1000 bootstrap sampling, Set 2 carried 89.8% chance of having a lower AIC value and therefore demonstrated itself as a better model in predicting OS (mean AIC 228.1, 95% CI: [194.8; 251.8]) compared to Set 1 (mean AIC 278.5, 95% CI: 254.6–301.2, *p* < 0.001).

### 3.4. External Validation of Anatomic-RPA Groups and Prognostic-RPA Groups in the Validation Cohort

The 3-year OS and the median OS were 29.4% and 28.0 months, respectively, in the validation cohort. MVA in the validation cohort confirmed that the co-existence of liver-bone metastases was the only significant OS prognostic factor in Set 1 (*p* = 0.017) and pre-treatment plasma EBV DNA was the only prognostic factor of OS in Set 2 (*p* = 0.008) (Table 2 and Table 4), consistent with the findings observed in the training cohort. When applying Set 1 and Set 2 stage groups in the validation cohort after RPA, significant survival differences among M1 categories and prognostic groups were still observed. The median OS of patients in M1a and M1b in Set 1 were 26.3 months and 9.8 months, respectively (*p* = 0.017; Figure 3b and Table 3). Similarly, statistical differences in the median OS were revealed between M1a and M1b in Set 2 (33.2 months and 15.9 months, respectively (*p* = 0.001) (Figure 4b and Table 5).

After 1000 bootstrapping replications in this validation cohort, Set 2 performed better in OS prediction, with the mean C-index of 0.59 (95% CI: 0.54–0.67), higher than that for Set 1 (0.47, 95% CI: 0.41–0.53, *p* < 0.001).

## 4. Discussion

The AJCC/UICC TNM framework is the internationally recognized standard to describe tumor extent, which helps treatment planning, prognostication, stratification into clinical trials, and treatment response evaluation [1,2,3,4,34,35]. Nonetheless, heterogeneity exists in the survival outcomes in patients with M1 disease and a subgroup of this patient population has shown long-term survival [5,6,7,8,9,10,11,12,13,14,15,16,17,18,19,20,21,36,37,38]. However, the current TNM-8 is not able to identify the survival differences among these NPC patients with de novo metastases. The catch-all denotation of the M1 category cannot accurately predict their prognosis. In view of this limitation, further to our recently published pooled analysis [22], we conducted this study in an attempt to subdivide M1 within the TNM framework to help better stratify M1 patients for further personalized treatment.

Although various previously designed prognostic models like nomograms and prognostic index score (PIS) systems might offer prognostic information on NPC [15,19,39,40,41,42,43], they are limited by their complexity and are initially designed for individual risk estimation but not for staging purposes. Meanwhile, other studies, in general, focused on ordinary radiological characteristics of distant metastases and were based on conventional UVA and MVA only [5,6,7,8,9,10,11,12,13,14,15,16,17,18,19,20,21]. On the contrary, this study, with the use of MVA and robust RPA subsequently, along with both internal and external validations, clearly demonstrated that anatomic factors could significantly segregate de novo M1 NPC, which derived from our Anatomic-RPA groups. These further validated the results in our recently published pooled analysis separating the M1 category into M1a and M1b by the co-existence of liver and bone metastases [22].

Non-anatomic factors have been recently introduced by the AJCC/UICC for further patient prognosis segregation while maintaining the conventional anatomical stage groups. To date, non-anatomic factors have been incorporated into the recent staging modifications for some other diseases, for example, breast cancer and esophageal cancer [44,45]. The role of plasma EBV DNA in NPC prognosis has been extensively studied and it is so far the most accurate biomarker for the management of NPC [23,24,25,26,27,28,29,30,31]. With respect to NPC, only two studies, including ours, demonstrated the superiority of survival prediction with the incorporation of such a non-anatomic factor into the TNM staging system over the TNM-8 stage groups for non-metastatic disease [33,46]. Two recently published retrospective studies in China attempted to subdivide the M1 category by plasma EBV DNA, but one of them was mainly for rM1 disease, while another was conducted without validation and during the time when IMRT was not fully implemented [47,48].

Our study is the first demonstrating that pre-treatment plasma EBV DNA, which trumped anatomic factors as the only prognostic factor of OS in another UVA and MVA in our study, was the most significant non-anatomic factor to further derive our prognostic-RPA groups. Here we revealed that pre-treatment plasma EBV DNA of 2500 copies/mL was a robust cutoff to subdivide M1 into two prognostic groups for de novo M1 NPC. We further proved that plasma EBV DNA alone was better than anatomic factors in predicting OS. This is in line with the recent consensus statement made by the panel members of AJCC and UICC that the original and fundamental purpose of staging is to provide a classification that reflects the anatomic extent of disease generalizable across the full spectrum of disease and geographical regions [34], while the amalgamation of other non-anatomic prognostic factors, i.e., plasma EBV DNA in our study described further below under the essential anatomic components within TNM provides further prognosticative, which might be more applicable in endemic regions where access to well-established and reliable biomarker assay platforms in high-volume treatment centers is available. It is also worthwhile in the future to explore dynamic changes of plasma EBV DNA, correlation with tissue EBV DNA and other potential biomarkers to increase the accuracy of patient stratification for more personalized treatment.

One may argue that the accuracy of plasma EBV DNA assay varies among institutions, which may be one of our study limitations, and our results may not be reproducible in other countries/regions in which a reliable, sensitive, and validated EBV DNA assay platform is not available. We acknowledge this potential pitfall though we measured plasma EBV DNA consistently within 4 h of blood sampling and used the same methodology in the same institution for all patients to avoid any inconsistency and error due to delayed processing, as reflected by the discrepancy of <2% of all samples in our internal validation as previously reported by us [33]. Our current study used the most accurate assay devised by Lo et al., of which the sensitivity has much improved and the lowest detectable limits reduced from 60 copies/mL to 20 copies/mL over the past 15 years [23,49]. Nevertheless, even if Lo’s assay is used, its sensitivity to diagnose NPC was only up to 97.1%, as shown in their NPC screening study in Hong Kong [32]. In our recent publication, on the other hand, our assay could detect plasma EBV DNA below 20 copies/mL in our prospective cohort of 518 NPC patients. In fact, 62 (12.0%) and 78 (15.1%) of our cohort with histologically confirmed previously untreated NPC had 0 copy and 1–20 copies/mL, respectively, of EBV DNA in their plasma at diagnosis and their survival was not better than their counterparts of the same stage [50]. It is indeed very difficult to achieve international harmonization among institutions because plasma EBV DNA assays are laboratory-derived tests. The diverse heterogeneity of assays among institutions, in terms of extraction and amplification technique, primers and probes targeting different regions of the EBV genome, the use of master mixes and calibrators, quantification controls, and reporting units for the results are the major hurdles as commented by Kim et al. [29,32,51]. Though harmonization significantly improved the sensitivity of plasma EBV DNA as conducted by Le et al. in their multicenter study, up to 17% of discordance of the titers could still be observed [32]. In addition, as revealed from the NPC screening study in Hong Kong, the sensitivity and specificity of plasma EBV DNA could be potentially affected by other patient and environmental factors [31,52]. While the above reasons can affect the performance of plasma EBV DNA assays, this is less than a concern to our study because all the patients had metastatic disease in which their tumor volumes are large enough to produce circulating tumor DNA. Only 1 patient had 20 copies/mL in the training cohort and 1 patient had 0 copy/mL in the validation cohort in this study. The reasons for extremely low plasma EBV DNA in NPC, even in endemic regions, have been explained by us recently [50].

The use of another group of initially non-metastatic NPC patients who later developed distant metastasis in our observational study as the validation cohort for external validation may be another study limitation. Similar cohorts comprising de novo M1 NPC from other institutions may serve as better validation cohorts for external validation. However, it should be noted that the lowest detection limit of EBV DNA assays among different institutions varies widely, resulting in variation in optimal cutoff levels. While our assay could detect plasma EBV DNA below 20 copies/mL, it is relatively difficult to include EBV DNA data from other institutions with higher detection limits as a validation unless international harmonization to improve standardization is achieved [29,31]. We should also realize that treatment paradigms for de novo M1 NPC are much more diversified and inconsistent among institutions, especially when it comes to the choice of initial chemotherapy and maybe, more importantly, the employment of consolidative IMRT and local ablative therapy with surgery and/or SBRT for oligometastases, which can only be achievable in high-volume centers with expertise and radiation resources.

Further studies are warranted to explore the most optimal treatment after risk stratification, while our study highlighted the importance of consolidative IMRT for non-progressive disease following initial chemotherapy and additional SBRT for oligometastases [53]. This truly reflects the modern real-world practice in high-volume centers equipped with contemporary radiation facilities and experts. Our results echoed the findings published in the recent phase III RCT which showed an OS and PFS improvement with consolidative radical IMRT following chemotherapy [54]. However, the use of an older chemotherapy regimen in that study may not reflect the most standard practice [54,55]. The results of another recently published retrospective study with a similar treatment paradigm of initial chemotherapy followed by IMRT were also in line with our findings [56]. In addition, retrospective studies in the United States and China revealed that locoregional radiation therapy improved OS, especially when radiation doses were ≥50 Gy and ≥65 Gy, respectively, but the exact timing between initial chemotherapy and locoregional radiation therapy was yet to define [57,58]. It is highly expected that more prospective studies could provide a clearer answer.

## 5. Conclusions

In conclusion, we proposed an M-category subdivision stratified by anatomic factors for de novo M1 NPC, which can be generally applied in different geographical regions. Further RPA-derived prognostic groupings with plasma EBV DNA provided a more accurate survival prediction, resulting in a more precise and subsequent personalized treatment strategy for high-risk patients. Validation of our proposed M1 subdivision with other institutions is highly awaited.

## Figures and Tables

**Figure 1 cancers-14-01923-f001:**
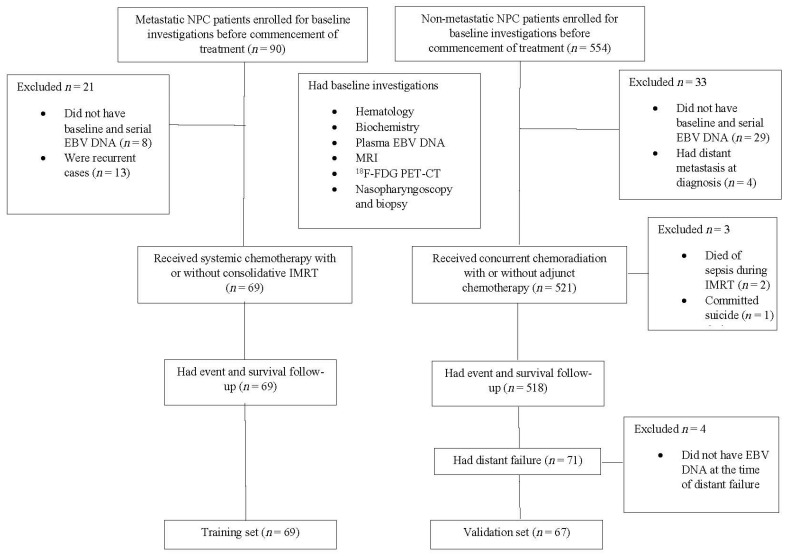
Study flowchart. NPC, nasopharyngeal carcinoma; EBV DNA, Epstein–Barr virus deoxyribonucleic acid; IMRT, intensity modulated radiation therapy; MRI, magnetic resonance imaging; ^18^F-FDG PET-CT, 18-Fluorodeoxyglucose positron emission tomography.

**Figure 2 cancers-14-01923-f002:**
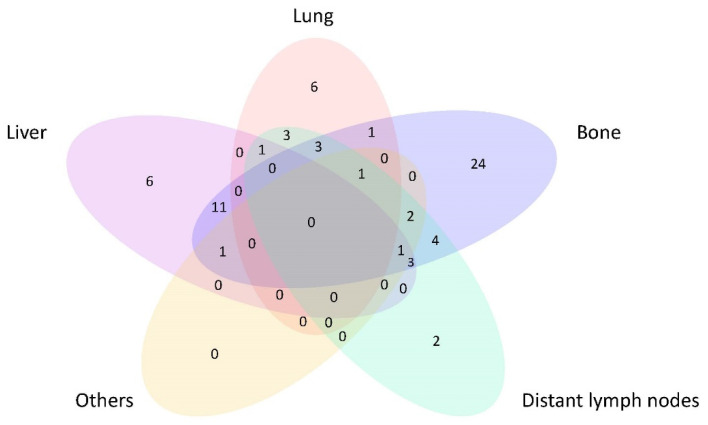
Venn diagram showing the distribution of various sites of distant metastases at presentation in the training cohort.

**Figure 3 cancers-14-01923-f003:**
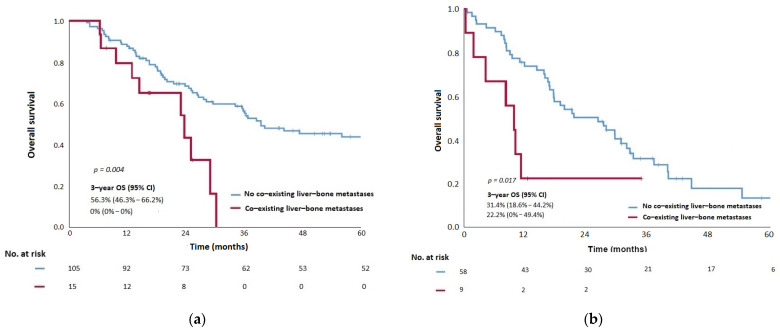
Kaplan–Meier curves showing (**a**) overall survival (OS) of patients of proposed M1a and M1b categories stratified by the presence of co-existing liver-bone metastases in Anatomic-RPA groups derived in Set 1 RPA in the training cohort, (**b**) OS of patients of proposed M1a and M1b categories stratified by the presence of co-existing liver-bone metastases in Anatomic-RPA groups derived in Set 1 RPA in the validation cohort. RPA, Recursive partitioning analysis.

**Figure 4 cancers-14-01923-f004:**
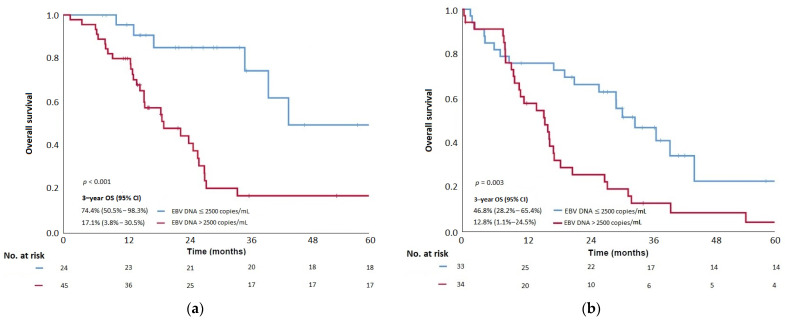
Kaplan–Meier curves showing (**a**) overall survival (OS) of patients of proposed M1a and M1b categories stratified by the pre-treatment plasma EBV DNA >2500 copies/mL in prognostic-RPA groups derived in Set 2 RPA in the training cohort, (**b**) OS of patients of proposed M1a and M1b categories stratified by the pre-treatment plasma EBV DNA >2500 copies/mL in prognostic-RPA groups derived in Set 2 RPA in the validation cohort. RPA, recursive partitioning analysis; EBV DNA, Epstein–Barr virus deoxyribonucleic acid.

**Table 1 cancers-14-01923-t001:** Baseline demographic and clinical characteristics.

	Training Cohort	Validation Cohort	*p*
Characteristic	No. of Patients (%)	No. of Patients (%)	
	Total (*n* = 69)	Total (*n* = 67)	
Median follow-up (months) (range)	17.8 (1.4–150.2)	17.7 (0.3–77.8)	0.589
Median (months)/3-year OS (%)	26.6 (30.0)	29.4 (28.0)	0.782
Median age in years (range)	55 (13–80)	49 (26–84)	0.180
Male/female	58 (84.1)/ 11 (15.9)	52 (77.6)/ 15 (22.4)	0.339
ECOG			0.025
0–1	64 (92.8)	67 (100)	0.025 0.193
2–3	5 (7.2)	0 (0)	
T-classification			
T1	9 (13.0)	11 (16.4)	0.193 0.697
T2	14 (20.3)	8 (11.9)	
T3	29 (42.0)	38 (56.7)	
T4	17 (24.6)	10 (14.9)	
N-classification			
N0	4 (5.8)	2 (3.0)	0.697
N1	8 (11.6)	8 (11.9)	
N2	25 (36.2)	30 (44.8)	
N3	32 (46.4)	27 (40.3)	
Median pre-treatment plasma EBV DNA in copies/milliliter at the time of distant metastasis diagnosed (range)	4919 (20–7,687,500)	2516 (0–2,100,000)	0.092
Lung metastasis			0.003
Present	15 (21.7)	31 (46.3)	0.003 <0.001
Absent	54 (78.3)	36 (53.7)	
Bone metastasis			
Present	51 (73.9)	28 (41.8)	<0.001 0.637
Absent	18 (26.1)	39 (58.2)	
Liver metastasis			
Present	23 (33.3)	25 (37.3)	0.637 0.266
Absent	46 (66.7)	42 (62.7)	
Distant nodal metastasis			
Present	20 (29.0)	26 (38.8)	0.266 0.025
Absent	49 (71.0)	41 (61.2)	
Other metastases	5 (7.2)	0 (0)	
Number of metastatic sites			0.873
1	38 (55.1)	35 (52.2)	
2	19 (27.5)	22 (32.8)	0.873
3	10 (14.5)	9 (13.4)	
4	2 (2.9)	1 (1.5)	

OS, overall survival; EBV DNA, Epstein–Barr virus deoxyribonucleic acid; ECOG, Eastern Cooperative Oncology Group.

**Table 2 cancers-14-01923-t002:** First-round univariable and multivariable analyses of demographic and anatomic variables prognostic of overall survival.

	Training Cohort				Validation Cohort			
	Univariable Analysis		Multivariable Analysis		Univariable Analysis		Multivariable Analysis	
	Hazard Ratio (95% CI)	*p*	Hazard Ratio (95% CI)	*p*	Hazard Ratio (95% CI)	*p*	Hazard Ratio (95% CI)	*p*
Age (every 1-year increment)	1.02 (0.99–1.05)	0.15	-	-	1.02 (0.99–1.05)	0.16	-	-
Male (vs. female)	1.83 (0.65–5.19)	0.26	-	-	1.46 (0.70–3.03)	0.31	-	-
T-category (3–4) vs. (1–2)	1.00 (0.51–1.97)	0.99	-	-	1.43 (0.73–2.80)	0.30	-	-
N-category (2–3) vs. (0–1)	1.99 (0.70–5.63)	0.20	-	-	1.40 (0.83–2.54)	0.20	-	-
Multiple metastatic sites (vs. single site)	1.45 (0.76–2.76)	0.26	-	-	1.45 (0.83–2.54)	0.20	-	-
Lung metastasis	0.96 (0.46–2.00)	0.92	-	-	0.75 (0.42–1.33)	0.32	-	-
Bone metastasis	1.30 (0.64–2.63)	0.47	-	-	1.45 (0.82–2.56)	0.20	-	-
Liver metastasis	1.50 (0.74–3.01)	0.26	-	-	1.58 (0.91–3.16)	0.21	-	-
Distant nodal metastasis	0.91 (0.46–1.81)	0.79	-	-	0.85 (0.47–1.51)	0.57	-	-
Other metastases	0.48 (0.11–2.00)	0.32	-	-	#	#	#	#
Co-existing liver-bone metastases	2.37 (1.10–5.11)	0.023 *	2.37 (1.10–5.11)	0.023 *	2.63 (1.15–6.01)	0.017 *	2.63 (1.15–6.01)	0.017 *
Co-existing bone-lung metastases	0.92 (0.28–3.01)	0.89	-	-	1.74 (0.77–3.93)	0.18	-	-
Co-existing liver-lung metastases	1.45 (0.20–10.70)	0.72	-	-	1.68 (0.78–3.62)	0.19	-	-
Co-existing bone-distant nodal metastases	0.71 (0.30–1.71)	0.49	-	-	1.47 (0.58–3.76)	0.42	-	-
Co-existing liver-distant nodal metastases	1.19 (0.69–2.05)	0.54	-	-	1.29 (0.51–3.26)	0.6	-	-
Co-existing lung-distant nodal metastases	1.14 (0.48–76)	0.76	-	-	0.81 (0.39–1.70)	0.58	-	-

95% CI, 95% confidence interval. * Only covariates found significant (*p* < 0.1) in the univariable analysis were considered in the multivariable analysis. Significant covariates are in bold. # No metastases in other parts of the body in the validation cohort.

**Table 3 cancers-14-01923-t003:** Overall survival of M1 segregation by the co-existence of liver and bone metastases in Set 1 RPA (Anatomic-RPA groups) in the training and validation cohorts.

	Training Cohort			Validation Cohort		
	No Co-Existing Liver-Bone Metastases (M1a)	Co-Existing Liver-Bone Metastases (M1b)	*p*	No Co-Existing Liver-Bone Metastases (M1a)	Co-Existing Liver-Bone Metastases (M1b)	*p*
Overall survival			0.023			0.017
3-year rate (95% CI)	41.8% (25.8–57.8%)	0% (0–0%)		31.4% (18.6–44.2%)	22.2% (0–49.4%)	
Median (months) (95% CI)	28.1 (18.6–37.5)	19.2 (11.7–26.6%)		26.3 (15.8– 36.8)	9.8 (5.1–14.6)	
Mean (months) (95% CI)	56.0 (37.2–74.8)	17.9 (13.2–22.7)		29.9 (23.2–36.6)	12.8 (4.8–20.8)	
HR (95% CI)	2.37 (1.10–5.11)			2.63 (1.15–6.01)		

95% CI, 95% confidence interval; HR, hazard ratio.

**Table 4 cancers-14-01923-t004:** Second-round univariable and multivariable analyses of prognostic variables of overall survival with the addition of pre-treatment plasma EBV DNA.

	Training Cohort				Validation Cohort			
	Univariable Analysis		Multivariable Analysis		Univariable Analysis		Multivariable Analysis	
	Hazard Ratio (95% CI)	*p*	Hazard Ratio (95% CI)	*p*	Hazard Ratio (95% CI)	*p*	Hazard Ratio (95% CI)	*p*
Age (every 1-year increment)	1.02 (0.99–1.05)	0.15	-	-	1.02 (0.99–1.05)	0.16	-	-
Male (vs. female)	1.83 (0.65–5.19)	0.26	**-**	**-**	1.46 (0.70–3.03)	0.31	**-**	**-**
T-classification (3–4) vs. (1–2)	1.00 (0.51–1.97)	0.99	**-**	**-**	1.43 (0.73–2.80)	0.30	**-**	**-**
N-classification (2–3) vs. (0–1)	1.99 (0.70–5.63)	0.20	**-**	**-**	1.40 (0.83–2.54)	0.20	**-**	**-**
Log_10_ Plasma EBV DNA	1.57 (1.20–2.04)	0.001 *****	1.46 (1.08–1.99)	0.015 *****	1.40 (1.11–1.75)	0.004 *****	1.38 (1.08–1.74)	0.008 *****
Multiple metastatic sites (vs. single site)	1.45 (0.76–2.76)	0.26	**-**	**-**	1.45 (0.83–2.54)	0.20	**-**	**-**
Lung metastasis	0.96 (0.46–2.00)	0.92	**-**	**-**	0.75 (0.42–1.33)	0.32	**-**	**-**
Bone metastasis	1.30 (0.64–2.63)	0.47	**-**	**-**	1.45 (0.82–2.56)	0.20	**-**	**-**
Liver metastasis	1.50 (0.74–3.01)	0.26	**-**	**-**	1.58 (0.91–3.16)	0.21	**-**	**-**
Distant nodal metastasis	0.91 (0.46–1.81)	0.79	**-**	**-**	0.85 (0.47–1.51)	0.57	**-**	**-**
Other metastases	0.48 (0.11–2.00)	0.32	**-**	**-**	**#**	**#**	**#**	**#**
Co-existing liver-bone metastases	2.37 (1.10–5.11)	0.027 *****	1.45 (0.59–3.56)	0.420	2.63 (1.15–6.01)	0.022 *****	2.18 (0.95–5.00)	0.066
Co-existing bone-lung metastases	0.92 (0.28–3.01)	0.89	**-**	**-**	1.74 (0.77–3.93)	0.18	**-**	**-**
Co-existing liver-lung metastases	1.45 (0.20–10.70)	0.72	**-**	**-**	1.68 (0.78–3.62)	0.19	**-**	**-**
Co-existing bone-distant nodal metastases	0.71 (0.30–1.71)	0.49	**-**	**-**	1.47 (0.58–3.76)	0.42	**-**	**-**
Co-existing liver-distant nodal metastases	1.19 (0.69–2.05)	0.54	**-**	**-**	1.29 (0.51–3.26)	0.6	**-**	**-**
Co-existing lung-distant nodal metastases	1.14 (0.48–76)	0.76	**-**	**-**	0.81 (0.39–1.70)	0.58	**-**	**-**

95% CI, 95% confidence interval; EBV DNA, Epstein–Barr virus deoxyribonucleic acid. * Only covariates found significant (*p* < 0.1) in the univariable analysis were considered in the multivariable analysis. Significant covariates are in bold. # No metastases in other parts of the body in the validation cohort.

**Table 5 cancers-14-01923-t005:** Overall survival of M1a (pre-treatment plasma EBV DNA ≤2500 copies/mL) and M1b (pre-treatment plasma EBV DNA >2500 copies/mL) derived by Set 2 RPA (prognostic-RPA groups) in training and validation cohorts.

	Training Cohort			Validation Cohort		
	Pre-Treatment Plasma EBV DNA ≤2500 Copies/mL (M1a)	Pre-Treatment Plasma EBV DNA >2500 Copies/mL (M1b)	*p*	Pre-Treatment Plasma EBV DNA ≤2500 Copies/mL (M1a)	Pre-Treatment Plasma EBV DNA >2500 Copies/mL (M1b)	*p*
Overall survival			<0.001			0.001
3-year rate (95% CI)	74.4% (50.5–98.3%)	17.1% (3.8–30.5%)		46.8% (28.2–65.4%)	12.8% (1.1–24.5%)	
Median (months) (95% CI)	44.2 (34.7–53.8)	19.7 (12.1–30.0)		33.2 (23.4–43.0)	15.9 (9.7–22.0)	
Mean (months) (95% CI)	66.7 (45.5–87.8)	32.7 (6.7–19.6)		34.3 (25.3–43.4)	19.9 (13.8–26.0)	
HR (95% CI)	4.38 (1.82–10.58)			2.16 (1.17–3.98)		

EBV DNA, Epstein–Barr virus deoxyribonucleic acid; HR, hazard ratio; 95% CI, 95% confidence interval; RPA, recursive partitioning analysis.

## Data Availability

The datasets generated for this study will not be made publicly available since national legislation and the terms of study ethics approval do not allow dataset sharing outside of the institutions participating in the analysis.

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
