# Peer review of "An Exploratory Study of Refining TNM-8 M1 Categories and Prognostic Subgroups Using Plasma EBV DNA for Previously Untreated De Novo Metastatic Nasopharyngeal Carcinoma"

_cancers, 2022, doi:10.3390/cancers14081923_

Round 1

Reviewer 1 Report

Thank you very much for this article. I look forward to seeing the validation studies in the near future. It would be very interesting to see if we can use your findings can also help us in setting up a more rationale surveillance programs after therapy.

Author Response

Please kindly refer to the attached file for the reply to Reviewer 1 comments.

Reviewer 2 Report

NPC patients with de novo distant metastasis appears to be a heterogeneous group who demonstrate a wide range of survival as suggested by growing evidence.

The Authors suggest to segregate survival for de novo M1 NPC by anatomic characteristics (lung, bone, liver metastasis etc) and pre-treatment plasma Epstein-Barr virus (EBV) DNA respectively.

The Authors proposed that this new division by anatomic factors and EBV DNA level may be applied in different part of the world and can be helpful for personalized therapy for NPC patients. 

Whether the topic is original? These studies seem interesting. However, the authors propose to change the existing TNM classification. According to the current classification, group M is divided into Mo (without distant metastases) and M1 (with distant metastases). The article is controversial as the authors propose a new division of the M1 group.

The Authors propose to divide this group (M1) into subgroups according to the anatomical site of the metastasis, e.g. lung, liver, etc. An additional criterion for the division would be the level of EBV DNA. The viral load was divided into two groups more than 2500 copies EBV DNA/mL and less than 2500 copies/mL.

My major comments are following:

The article is written too broadly. The way of presenting it is hardly understandable to the reader. Authors should rewrite the article, i.e.

  1. The work should clearly specify what is the subject of the analysis
  2. Then what is the aim of this study
  3. detailed characteristics of the studied patients
  4. detailed description of the methods used in this study
  5. justification why the introduction of the new classification is important and what is its clinical significance
  6. comments on methods:

a/ First of all, the method of EBV DNA determination should be precisely described, and not only a reference to previous publications, therefore the authors propose changes in the current classification. Therefore, the breakdown criteria should be clearly defined.

b/The question arises why, according to the authors, they took into account the division into below and above 2500 copies/mL of DNA?

c/the group of 69 patients is too small to draw far-reaching conclusions. It can only be a preliminary study.

d/ The authors only assessed plasma EBV DNA. Why did they not assess the presence of EBV DNA in the tissue? I propose to determine the presence of EBV DNA in the tissue and show that it correlates with the presence of DNA in the plasma.

e/ Table 2 must contain an exact legend; it should be noted what is statistically significant;

the viral load should be recorded correctly, e.g. 2.5 x 103 copies/mL

f/ all abbreviations used in the text and in tables must be explained in parentheses. The work must be written transparently for the reader.

In conclusion, the article requires a thorough redrafting before publication. In addition, I propose to change the title to include that these are preliminary studies with a proposal for an additional metastases classification. 

Author Response

Please kindly refer to the attached file for the reply to Reviewer 2 comments.

Reviewer 3 Report

The authors performed an prospective analysis of anatomic prognostic factors for de novo M1 NPC.  They demonstrated that co-existing liver-bone metastases was the only prognostic factor, which can be generally applied in different geographical regions. Addition of pre-treatment plasma EBV DNA to UVA/MVA and further RPA-derived prognostic groupings with plasma EBV DNA provided a more accurate survival prediction, which can lead to more precise and subsequent personalized treatment strategy for high-risk patients. Surprisingly, the authors validated their results well using other cohort of rM1 NPC. Their report is really interesting and can impact on head and neck oncologists.

Some minor issues should be reconsidered.

Line 30, it is the first showing up of RPA in this report.

Line 198, "from the training cohort" is repeated.

Line 292, they should be Supplementary Figure "S"2a and "S"2b.

Line 59, 293-, The median OS of M1a and M1b in Table 5 are shown as "Not reached and 24.6 months". Is "44.2 months vs 19.7 months" derived from Figure 4a?

 Line 321, p=0.001 is described as p=0.003 in Table 5.

Line 360, Does it mean "consolidation IMRT following initial chemotherapy"?

Line 405-406, this sentence needs a word "respectively".

Author Response

Please kindly refer to the attached file for Reviewer 3 comments.

Round 2

Reviewer 2 Report

The Authors modified the study title because the relatively small sample size is one of the study limitations. 

The Authors revised your manuscript according to reviewers' comments and suggestions. I think, the new form of paper may be accepted for publication.